# Hooked and Distracted? A Network Analysis on the Interplay of Social Media Addiction, Fear of Missing Out, Cyberloafing, Work Engagement and Organizational Commitment

**DOI:** 10.3390/bs15121719

**Published:** 2025-12-11

**Authors:** Phillip Ozimek, Anna Sander, Nele Borgert, Elke Rohmann, Hans-Werner Bierhoff

**Affiliations:** 1Department of Humanities, Vinzenz Pallotti University, 56179 Vallendar, Germany; 2Department of Social Psychology, Faculty of Psychology, Ruhr University Bochum, 44801 Bochum, Germany; 3Institute of Psychology, University of Bern, 3012 Bern, Switzerland

**Keywords:** social media addiction, Fear of Missing Out, cyberloafing, work engagement, organizational commitment

## Abstract

This study investigates interrelations among social media addiction (SMA), Fear of Missing Out (FoMO), cyberloafing (CL), work engagement (WE), and organizational commitment (OC) using network analysis. An online survey in Germany/Switzerland (n = 452; OC assessed in the employed subsample, n = 173) measured the five constructs. Unregularized and EBICglasso partial-correlation networks were estimated, and centrality and bridge indices were computed. Two robust edges emerged: a strong SMA–FoMO association and a strong positive WE–OC link; the regularized network additionally indicated a triangular SMA–FoMO–CL pattern. FoMO and OC acted as bridge nodes between problematic social media behaviors and work attitudes, whereas direct SMA links to WE/OC were weak or absent. Findings position FoMO as a pivotal mechanism connecting social media use to organizational attitudes and support, distinguishing functional micro-breaks from disruptive CL. Limitations include the cross-sectional design, student-skewed sample, self-report measures, smaller OC subsample, and a German/Swiss context.

## 1. Introduction

Private social media use during work is pervasive and yields mixed outcomes: it can facilitate communication and brief recovery, but it may also disrupt attention and performance ([38]; [42]; [119]). A central gap is that prior research rarely examines social media addiction, Fear of Missing Out (FoMO), cyberloafing, work engagement, and organizational commitment within a single model, leaving their interdependencies and potential bridging mechanisms unclear.

Social media addiction denotes excessive, compulsive use consistent with core addiction features ([7]; [43]). Platform design and social rewards can reinforce frequent checking and sustained use ([65]; [115]; [52]). FoMO is the persistent concern about missing rewarding experiences; it relates to frustrated psychological needs and shows cyclical amplification with social media use ([93]; [18]; [99]). In work contexts, such vigilance can manifest as cyberloafing—nonwork online activity during work hours—with minor versus more serious forms distinguished ([70]; [15]; [40]). Work engagement reflects vigor, dedication, and absorption ([105]), while organizational commitment—here, affective commitment—captures emotional attachment and identification with the organization ([83]; [4]).

Using network analysis, we map joint associations, identify robust edges, and detect bridge nodes between problematic social media behaviors and work attitudes ([31]; [32]; [57]; [101]). We expect strong links between social media addiction and FoMO and between work engagement and organizational commitment, positive connections from social media addiction and FoMO to cyberloafing, and weaker or absent direct links from social media addiction to work engagement and organizational commitment once other variables are considered ([20]; [113]; [21]; [114]; [46]; [11]; [3]).

## 2. Theoretical Background

### 2.1. Social Media Use

Social networks are internet-based platforms that allow users to create personal profiles and maintain or build relationships with others, both offline acquaintances and new contacts ([30]). Key elements include internet connectivity and real-time or asynchronous communication ([19]). Due to the continuous availability of internet-enabled smartphones and other mobile devices, social media usage became a core online activity in recent years ([86]). The most common reasons for using social media are social interaction, information seeking, pastime, entertainment, relaxation, the facilitation of communication, and user convenience ([123]).

#### 2.1.1. Social Media Addiction

Social media addiction is defined as the excessive use of social networks involving a significant investment of time and energy, which is potentially harming both professional and personal life (e.g., [111]; [7]). The nature of social media itself can facilitate compulsive behavior and potentially foster addiction ([125]).

According to [43] ([43]), addiction is defined by six criteria: salience, mood modification, tolerance, withdrawal, conflict, and relapse. Social media addiction includes prioritizing usage over other activities (salience), emotional shifts (mood modification), increased use over time (tolerance), irritability when access is reduced (withdrawal), interference with other life areas (conflict), and returning to excessive use after attempts to quit (relapse; [43]).

Although the terms *social media addiction*, *problematic social media use* and *compulsive social media use* are often used interchangeably, they carry slightly different connotations and therefore reflect a different understanding of excessive social media use ([13]).

Younger people seem more at risk of developing addiction symptoms yet often perceive their usage as unproblematic and part of daily life ([66]).

#### 2.1.2. Motivational Drivers of Social Media Addiction

Social media platforms, such as Facebook or Instagram, reinforce users to increasingly associate app use with frequent social rewards ([52]). Due to this reinforcement mechanism, checking behavior increases because users seek gratifying notifications ([52]). Combined with the minimal effort required to check one’s social media accounts, this mechanism creates a strong incentive for repeated and potentially addictive use ([52]; [75]). The benefits of using social media may instigate people to use social networks excessively and habitually, which in turn is likely to foster the development of addictive behaviors ([115]).

### 2.2. Fear of Missing Out

FoMO describes the constant worry that other people experience rewarding experiences while one is absent ([93]). People who experience FoMO feel a persistent urge to remain constantly engaged with what is going on ([93]).

Social media platforms enable their users to share personal characteristics, such as successes and emotions, through pictures and other posts ([120]). This enables users to make upward social comparisons, comparing themselves with others perceived as more popular or socially engaged ([120]). These comparisons can be intensified by the selective emphasis on the most appealing parts of people’s lives in online self-presentation ([74]). Consequently, individuals who tend to compare themselves more frequently are more often exposed to others’ self-reported experiences on social media, which are often positively biased and focused on rewarding experiences ([99]). This, in turn, may foster the perception that others are doing better, thereby intensifying feelings of missing out ([99]).

#### 2.2.1. Different Contextual Conceptualizations of FoMO

When exploring Fear of Missing Out in workplace settings, researchers have generally taken two distinct approaches (e.g., [100]). In addition to the more general concept of FoMO, the concept of workplace FoMO was introduced by [17] ([17]) which describes the constant concern about missing out on rewarding experiences in the workplace context. [17] ([17]) propose that workplace FoMO consists of two key components: relational exclusion and informational exclusion.

#### 2.2.2. Cognitive and Behavioral Antecedents of Fear of Missing Out

Research on FoMO’s development focuses on its underlying psychological mechanisms, primarily linking it to the Self-Determination Theory (SDT); [26] ([26]); [93] ([93]). Situational or chronic deficits in psychological need satisfactions may constitute a risk factor for developing FoMO. Empirical studies have shown an association between low psychological need satisfaction and high levels of FoMO ([93]; [124]). Research has also found evidence for a reversed causal effect, whereby FoMO itself may negatively impact the satisfaction of psychological needs ([45]).

Furthermore, there is evidence that the use of social networking sites leads to a vicious cycle when experiencing FoMO ([18]; [93]). The experience of FoMO is often associated with decreased well-being ([18]). In an attempt to improve their well-being, people may tend to use social media more, which in turn leads to even stronger feelings of FoMO ([18]). In their longitudinal analysis, [18] ([18]) found empirical support for this cyclical relationship between social media use and the experience of FoMO.

### 2.3. Procrastination and Cyberloafing

#### 2.3.1. Procrastination

Procrastination refers to the voluntary postponement of a task due to a lack of motivation to complete it within the designated time ([107]). Delays typically occur when individuals substitute unpleasant or less attractive tasks with more appealing tasks, even while knowing this will likely lead to negative consequences. Chronic procrastinators often feel distress and recognize their behavior as irrational ([107]; [117]).

Procrastination is mostly described as a trait or disposition ([117]). Trait procrastination (chronic procrastination) is considered problematic and self-defeating because it impedes task completion ([33]). Nevertheless, procrastination may also be understood as a temporary state or process ([117]). Furthermore, [33] ([33]) distinguishes between functional and dysfunctional procrastination based on its outcome. Functional procrastination refers to situations in which one postpones a task because of prioritizing a more urgent one or to wait for crucial information that facilitates task completion ([33]). In contrast, dysfunctional procrastination refers to the maladaptive and inappropriate postponement of tasks ([34]).

#### 2.3.2. Cyberloafing

A subtype of procrastination that has received increasing attention in research is cyberloafing. According to [41] ([41]), cyberloafing is defined as nonwork-related use of a computer during working hours, which is unauthorized ([40]). [15] ([15]) differentiate between two types of cyberloafing. Minor cyberloafing is defined as a low-severity violation, such as sending and receiving private emails, online shopping or browsing the internet ([15]). Serious cyberloafing, on the other hand, constitutes more severe violations and includes behaviors such as using social media, communicating with others or engaging in online games.

#### 2.3.3. Psychological and Situational Causes of Cyberloafing

Cyberloafing at work is a failure in self-regulation, which, in turn, leads to a voluntary postponement of task completion despite being aware of its potential negative consequences ([79]). In research, procrastination is generally described as a strategy for evading unpleasant and undesirable tasks (avoidance procrastination), such as tasks perceived as boring ([109]).

Therefore, it can be argued that cyberloafing is a reaction to negative emotions in the workplace, which may be caused by dissatisfaction, stress and injustice, and can even lead to behaviors that harm the organization ([40]). Nevertheless, for most private internet use at work, it is more likely that individuals are attracted by the opportunities the internet offers than motivated by the desire to harm their employer ([40]). There is also some evidence for the existence of arousal procrastination, a type of procrastination driven by the urge to seek stimulation ([109]). Arousal procrastination is based on arousal-related theories of personality, such as sensation seeking ([109]).

### 2.4. Work Engagement

Work engagement is a positive psychological state characterized by employees being full of energy and being capable of dealing with their job demands ([105]). It reflects the extent to which employees fully apply their skills and potential when facing challenges at work. It is a motivational construct where employees feel an internal drive, strong focus, and sense of personal responsibility ([12]). When experiencing high levels of work engagement, employees feel an internal drive to overcome challenges and are willing to invest their full energy into the successful completion of tasks. Employees are highly focused, assume a strong sense of personal responsibility, and frequently experience a state of flow ([12]).

The term work engagement refers to three core dimensions: vigor, dedication, and absorption ([105]). Vigor refers to employees who are committed to their work with energy, persevere in the face of difficulties, and are mentally resilient (e.g., [108]). Dedication is characterized by feelings such as enthusiasm, passion, excitement, and pride (e.g., [108]). Absorption, finally, is characterized by a deep immersion and full concentration on one’s work (e.g., [108]). Typically, time seems to pass quickly, and one may struggle to detach from work (e.g., [108]). According to [105] ([105]), engagement is a pervasive and enduring cognitive state that is not directed toward any specific event, person or situation.

Note that the term work engagement is often associated with other related constructs, which can be conceptually distinguished from engagement. Work engagement is often regarded as the opposite of burnout. However, these two concepts are not perfectly negatively correlated ([105]). It is conceptually distinct from flow because engagement involves conscious self-regulation, whereas flow is characterized by unconscious immersion ([76]). Furthermore, work engagement is also conceptually distinguished from other attitudinal constructs such as job satisfaction or organizational commitment due to its strong link with task-related motivation ([23]).

### 2.5. Organizational Commitment

[4] ([4]) proposed a three-component model of commitment: affective commitment, normative commitment, and continuance commitment. Affective commitment describes the extent to which employees experience emotional attachment to the organization, identify with its values, and participate in the organization ([4]). In line with the definition by [83] ([83]), affective commitment is defined by three key components: belief in and acceptance of organizational aims and values, willingness to actively support and contribute to the organization, and a strong desire to remain a part of the organization.

Normative commitment refers to employees’ perceived obligation and sense of responsibility to remain with their organization ([4]). These obligations can result, for example, from regular payment or the unwillingness to weaken one’s own organization when it is facing financial difficulties ([58]).

Finally, continuance commitment refers to the potential financial, social, or psychological losses resulting from leaving the organization ([4]). When the perceived costs of leaving the organization (such as relocation, lower salary, or losses of benefits) are high, continuance commitment tends to increase. Furthermore, prior investments made within the organization and the resulting benefits, such as professional development or pension entitlements, further contribute to strengthening continuance commitment ([58]).

Furthermore, organizational commitment is different from work engagement, which relates to the employee’s experience with their work ([105]), whereas commitment reflects their overall loyalty and attachment to the organization itself ([83]).

### 2.6. Hypotheses and Research Question

On the basis of the theoretical background, six hypotheses and a research question were derived. First, we will formulate hypotheses about the relationships of SMA, FoMO, Cyberloafing, Work Engagement and Commitment. Second, the research question will discuss the theoretical considerations regarding the relationships between the constructs in a network.

#### 2.6.1. Social Media Addiction and Work Engagement

Social media addiction is assumed to have a negative association with work engagement. When employees are frequently distracted by social media, they are likely to experience interruptions while working on their regular tasks ([64]). Such behaviors impair their effort to spend full energy and concentration on their work, which may, in turn, reduce work engagement. This assumption is supported by research. For instance, [62] ([62]) found a significant negative relationship between social network site addiction and work engagement in a sample of Chinese employees. Similarly, [56] ([56]) revealed a negative association between social media addiction and employee engagement. In line with these findings, [54] ([54]) examined the relationship between social media addiction and work engagement among nurses, and their results confirmed a significant negative association. Therefore, a negative association between social media addiction and work engagement is proposed.

**H1.** 
*Social media addiction is negatively associated with work engagement.*


#### 2.6.2. Social Media Addiction and Organizational Commitment

People can be attached to social media, which means there is a bond between a person and social media ([118]). Social media can serve as a new attachment focus ([118]). Given its central role in facilitating connections with individuals, organizations, and brands, social media may offer users a sense of comfort, safety, and security similar to traditional relational attachments ([118]). Considering that social media offers continuous social gratification ([65]) and emotional reinforcement ([122]), it may become a more immediate and personally rewarding attachment focus than the workplace. Accordingly, we assume that individuals addicted to social media may reduce the importance of organizational commitment. This negative association between social media addiction and organizational commitment was already found in previous research ([22]).

**H2.** 
*Social media addiction is negatively associated with organizational commitment.*


#### 2.6.3. Social Media Addiction and Fear of Missing Out

Social network sites may be especially attractive for those with high levels of FoMO because of the possibility to communicate and stay in touch with others ([93]; [30]). Having high levels of FoMO and being unable to log in and browse through social media may lead to impulsive checking habits which, in turn, may result in an addiction ([44]). Higher levels of FoMO have been found to lead to more frequent checking of Facebook, Twitter, Instagram, and MySpace ([1]). The association between FoMO and Social Media Addiction has been confirmed by various studies (e.g., [20]; [113]). In line with previous findings, [39] ([39]) demonstrated that FoMO and social media addiction are significantly positively related.

**H3.** 
*Social media addiction is positively associated with Fear of Missing Out.*


#### 2.6.4. Fear of Missing Out and Cyberloafing

FoMO is associated with behaviors such as frequent social media use in daily life ([93]) and repetitive checking behaviors ([44]). Moreover, among students with high levels of FoMO it is common to use social media during lectures ([93]). These behaviors inhibit to work with concentration and therefore may foster procrastination. To support this, [121] ([121]), for example, identified a significant positive relationship between FoMO and procrastination. In the study by [21] ([21]), similar results were obtained concerning cyberloafing. FoMO had a direct effect on cyberloafing and was even more influential than personality traits ([21]). Interview participants reported engaging in cyberloafing due to FoMO ([21]). Accordingly, FoMO appears to be a key factor in individuals’ decisions to procrastinate ([98]). Also, from an SDT perspective (cf., [26]), FoMO indexes need frustration that motivates vigilant, frequent checking, which in work contexts may surface as cyberloafing.

**H4.** 
*Fear of Missing Out is positively associated with cyberloafing.*


#### 2.6.5. Cyberloafing and Work Engagement

Cyberloafing is believed to have a negative relationship with work engagement. Employees with high levels of work engagement are resilient, capable of coping with difficulties, and full of energy while working ([105]). Therefore, engaged employees may not feel the need to engage in non-work-related behaviors ([78]). In contrast, employees with low levels of work engagement may lack cognitive and physical stimulation from their work and are therefore more likely to seek enjoyable distractions, which may lead them to engage in cyberloafing behaviors ([78]).

**H5.** 
*Cyberloafing is negatively associated with work engagement.*


#### 2.6.6. Cyberloafing and Organizational Commitment

Organizational commitment is defined as an individual’s emotional attachment to, identification with, and involvement in the organization, as well as a perceived obligation to remain ([4]). In contrast, cyberloafing involves the use of the internet during working hours for personal purposes, which can reduce productivity and harm the organization ([81]; [84]). Such behavior reflects a lack of alignment with organizational goals and values and stands in contrast to the characteristics of strong commitment. Employees with low organizational commitment are less likely to feel morally obliged to use their time productively or act in the best interest of the organization, which makes them more prone to cyberloafing. This negative association has been supported by research (e.g., [53]; [87]).

**H6.** 
*Cyberloafing is negatively associated with Organizational Commitment.*


### 2.7. Research Question

When individuals experience social media addiction or Fear of Missing Out (FoMO), they often spend substantial time and energy engaging with social media, thereby depleting resources that would otherwise be available for work-related tasks ([61]; [125]). Cyberloafing, in this context, may serve as a short-term strategy to protect or regain resources ([63]), but over time, this behavior can lead to a loss of productivity and further resource depletion. These behaviors are associated with increased workplace social media use, which disrupts workflow ([89]). Such interruptions and distractions have been linked to increased time pressure, a higher probability of error, and reduced residual time for focused work ([10]; [77]).

Over time, resource-draining behaviors such as social media addiction and cyberloafing, as well as underlying psychological drivers like FoMO, may reduce the cognitive and emotional capacity needed to remain engaged at work and committed to the organization.

In sum, integrating social media addiction, FoMO, cyberloafing, work engagement, and organizational commitment into the framework of the COR theory provides a theoretically grounded approach to understand how these constructs may be interrelated through shared underlying dynamics. However, to date, no empirical study has simultaneously examined all five variables within a unified model, highlighting the need for an exploratory investigation such as the present network analysis. The Network is graphical depicted (cf., Section 4). This leads to the following research question:

Research Question: How are social media addiction, Fear of Missing Out (FoMO), cyberloafing, work engagement, and organizational commitment interrelated in a partial-correlation network, and which nodes bridge problematic social media behaviors and work-related attitudes?

We regard edges that persist in the regularized EBICglasso solution as comparatively robust; all network results are exploratory and non-causal.

## 3. Methods

### 3.1. Open Science Practices

This study was preregistered prior to 24 April 2025 on OSF: https://osf.io/um7wc/overview, accessed on 1 December 2025. We provide all our materials, analysis code, and data publicly accessible on OSF: https://osf.io/um7wc/resources, accessed on 1 December 2025. Due to the double-blind peer review, we will share the link after acceptance of the paper.

### 3.2. Sample and Procedure

A quantitative study was conducted using an online survey to investigate the proposed hypotheses and research question. Eligible to participate were people between 18 and 67 years of age who were currently studying or employed. Students working part-time alongside their studies were also eligible to participate. Another requirement for participation was being registered on at least one social media platform. Having an account on an instant messaging service such as WhatsApp or Signal was also sufficient, because social networking sites and instant messaging services share similarities and fulfil similar needs ([82]; [95]).

The result of the a priori power analysis using the statistical software R (Version 4.5.0 for Windows) and the pwr package indicated that a sample size of approximately 193 participants was appropriate. Note that the calculation was based on a significant level of α = 0.05, a desired power of 0.80, and an expected size of *r* = 0.20.

The study was conducted at the Ruhr University Bochum and the University of Bern. At the beginning of the survey, participants were asked to provide demographic data including gender, age, education, and employment status. Additionally, participants provided information about which social media platforms they use and how frequently they use them. Finally, participants were asked to look up their general screen time and their social media screen time on their smartphones. After completing the initial part of the study, various questionnaires were presented in randomized order (see Section 3.2, Measures). Since the sample primarily consisted of students, participants were allowed to decide whether they referred to their studies or part-time jobs when answering the work-related questionnaires. However, the questionnaire on organizational commitment could only be answered in relation to employment, as it was not applicable to the academic context. Completion of the survey took approximately 20 min.

In total, 506 individuals participated in the study. No participant was unemployed or retired. The data were checked for plausibility by excluding participants with more than 50% missing values or fewer than 25 valid responses. Additionally, participants who did not complete the main part of the survey were excluded. No uniform response patterns (e.g., repeatedly selecting “1”) were detected. In total, 54 participants were excluded. Consequently, the final sample of the study included 452 participants.

Specifically, the final sample consisted of 370 women (81.86%) and 80 men (17.70%). Two individuals identified as non-binary (0.44%). The sample primarily included younger participants with a mean age of 23.23 years (*SD* = 5.89). Because the survey was conducted in Germany and Switzerland, 40.27% (n = 182) of the participants came from Germany and 58.85% (n = 266) from Switzerland. A total of 90.27% (n = 408) of participants reported German as their mother tongue.

Most participants held the German/Swiss university entrance qualification (or equivalent) as their highest level of education (80.53%, n = 364). The second-largest group is people who had already completed a university degree (11.50%, n = 52). Correspondingly, the largest proportion of the sample consists of students (61.50%, n = 278) or students who work alongside their studies (35.84%, n = 162). Only 2.65% (n = 12) of participants reported being employed as their primary activity.

Additionally, participants were asked how many hours each week they dedicated to their studies and, if applicable, how many hours they worked in their job. Most students in the sample indicated working 21–30 h per week on their studies (28.64%, n = 126), followed closely by those working 31–40 h (27.27%, n = 120) and 11–20 h (22.73%, n = 100). A smaller number of participants reported spending 0–10 h per week on their studies (13.86%, n = 61), while only 6.59% (n = 29) reported working 41–50 h, and 0.91% (n = 4) reported more than 50 h per week. Regarding employment, 38.51% (n = 67) of participants reported working 0–10 h per week, 35.06% (n = 61) worked 11–20 h and 15.52% (n = 27) worked 21–30 h. Furthermore, 6.90% (n = 12) worked 31–40 h, and 4.02% (n = 7) reported working 41–50 h per week. For a more detailed overview of the demographic data, see Appendix A.

### 3.3. Measures

The following measurement instruments were used: the German version of the Bergen Social Media Addiction Scale (BSMAS), the German version of the Fear of Missing Out Scale, the German version of the Cyberloafing Scale, the German version of the Utrecht Work Engagement Scale (UWES), and the German version of the Organizational Commitment Questionnaire (OCQ). Appendix B provides a full list of the scales and items employed.

#### 3.3.1. Bergen Social Media Addiction Scale (BSMAS)

The Bergen Social Media Addiction Scale consists of six items based on the core criteria for an addiction ([6]). Responses are given on a 5-point Likert scale from “Very rarely” (1) to “Very often” (5). The items refer to the last year, for example, “How often during the last year have you felt an urge to use social media more and more?”. The BSMAS was adapted from the Bergen Facebook Addiction Scale (BFAS) by [8] ([8]) by replacing the word Facebook with the term social media. In the instructions, social media is described as “Facebook, Twitter, Instagram, and the like”. The German version of the BSMAS was validated by [16] ([16]). A composite mean score was computed, with higher values indicating more social media addiction. The internal consistency of the six-item scale in the present study was good, Cronbach’s α = 0.79.

#### 3.3.2. Fear of Missing Out Scale

The Fear of Missing Out Scale by [93] ([93]) consists of 10 items answered on a 5-point Likert scale ranging from “Not at all true of me” (1) to “Extremely true of me” (5). An example item is “I fear my friends have more rewarding experiences than me.” A composite mean score was computed, with higher values indicating higher levels of FoMO. The German version of the scalewas developed by [110] ([110]). It demonstrated an acceptable internal consistency in the present study (α = 0.74).

#### 3.3.3. Cyberloafing Scale

The Cyberloafing Scale that was used in this study was originally developed by [70] ([70]) andextended by [9] ([9]). As no validated German version of the questionnaire was available, the items were self-translated for the purpose of this study. The translation procedure is described in detail in Appendix C. The questionnaire consists of 19 items that describe various cyberloafing behaviors, such as “Visit sports related websites” and “Visit video sharing sites (YouTube, etc.)”. Participants are asked to rate on a 6-point Likert scale (“Never” (1)–“Once a day” (4)–“Constantly” (6) how often they engage in each of the behaviors. A composite mean score was computed with higher values indicating higher levels of cyberloafing. The internal consistency of the Cyberloafing Scale in the present study was very good (α = 0.89).

#### 3.3.4. Utrecht Work Engagement Scale (UWES)

The Utrecht Work Engagement Scale was developed by [106] ([106]) and consists of the three subscales vigor, dedication, and absorption. The original version consists of 17 items; the shortened version, which was used in the present study, consists of nine items, three for each subscale. Example items include “At my job, I feel strong and vigorous.” (vigor), “I am enthusiastic about my job.” (dedication), and “I am immersed in my work.” (absorption). In this study, only the general UWES score was used. Ratings take place on a 7-point Likert scale from “Never” (0) to “Always” (6). The German version of the UWES was developed and validated by [106] ([106]). A composite mean score was computed, with higher values indicating higher levels of work engagement. In the present study, the internal consistency of the UWES-9 scale was very good, α = 0.92.

#### 3.3.5. Organizational Commitment Questionnaire (OCQ)

The Organizational Commitment Questionnaire (OCQ) was first developed by [83] ([83]). The German version of the OCQ was validated by [58] ([58]) and consists of 15 items concerning affective commitment. The items are answered on a 5-point Likert scale ranging from “Totally disagree” (1) to “Totally agree” (5). Example items include “I would accept almost any type of job assignment in order to keep working for this company.” and “I feel very little loyalty to this organization.” A composite mean score was computed, with higher values indicating higher levels of organizational commitment. In the present study, the internal consistency of the Organizational Commitment Questionnaire was very good (α = 0.91). Organizational Commitment (OC) was only answered by participants who were currently employed. Thus, analyses involving OC are based on a reduced sample of employed participants (n = 173). The network was estimated using pairwise deletion, meaning that each edge was computed using all available cases for the respective pair of variables. As a result, edges involving OC are based on fewer observations than the rest of the network.

### 3.4. Statistical Analyses

Descriptive statistics and correlation analyses were performed using the *psych* package in R (version 4.5.0). To explore the bivariate relationships between key study variables, Pearson correlations were computed using pairwise deletion for missing values.

A network model involving the five measured variables, social media addiction, Fear of Missing Out, cyberloafing, work engagement, and organizational commitment was estimated using the *qgraph* and *igraph* packages in R. For estimating the network, pairwise partial correlations were used, which statistically controlled for the influence of the other variables in the network ([31]). The network was estimated using the pcor argument of the estimate Network function, which computes partial correlations using pairwise deletion. Thus, each edge was based on all available cases for the respective pair of variables, resulting in different effective sample sizes across edges.

To assess the importance of individual nodes, centrality indices were computed ([25]). Node strength measures the direct connectivity of a node to other nodes, closeness measures the indirect connectivity to other nodes, and betweenness captures how often a node appears on the shortest path linking two other nodes ([31]). Expected influence (one-step) corresponds node strength except it retains the positive or negative direction of each edge weight ([101]). To identify bridge nodes connecting social media-related variables and work-related attitudes, bridge strength and bridge expected influence (BEI) were calculated using the *networktools* package in R. Bridge strength is defined as the sum of all absolute edge weights, ignoring the direction ([57]). Higher values indicate more influence on the other community. Bridge expected influence is also a measure of the sum of all edge weights, but taking the direction into account ([57]).

The robustness of the network was evaluated using the *bootnet* package in R ([31]). First, bootstrapped 95% confidence intervals with 1000 bootstrap samples were calculated to assess the accuracy of the edge weights. A narrower confidence interval indicates a more reliable network ([85]). The stability of the centrality measures, strength, betweenness, closeness, and expected influence, was then estimated by calculating the correlation stability (CS) coefficient via a case-dropping bootstrap approach with 1000 bootstrap samples. The CS coefficient was recommended to be above 0.50 and never below 0.25 for sufficient stability ([31]).

To provide a comprehensive understanding of the network structure, two complementary network models were estimated. First, an unregularized partial correlation network was computed to offer an exploratory depiction of all potential associations among the variables. This approach displays the full pattern of relationships, including weaker connections that may not be robust. In contrast, the EBICglasso model applies regularization and model selection to remove potentially spurious or unstable edges and to shrink small partial correlations toward zero ([32]). As a result, the EBICglasso network typically contains fewer edges and highlights only the most reliable associations. Estimating both models allows for evaluating the robustness of the observed patterns.

## 4. Results

### 4.1. Social Media Use

Participants were asked to indicate which social media platforms they use. Multiple responses were allowed. The platforms used most frequently were WhatsApp (98.23%, n = 444) and Instagram (90.04%, n = 407). Other commonly used platforms included YouTube (78.10%, n = 353), Snapchat (61.95%, n = 280), and Pinterest (51.99%, n = 235). In contrast, platforms such as Telegram (7.96%, n = 36), Twitter/X (6.19%, n = 28), and XING (1.33%, n = 6) were used less frequently. The majority of participants reported using five (25.00%, n = 113), four (23.20%, n = 105), or six (21.00%, n = 95) different social media platforms. Only 2.21% (n = 10) of the respondents indicated using just one social media platform. Fewer than 5% (n = 19) of the respondents indicated using eight or more different social networks. A complete overview of the social media platforms used can be found in Appendix D.

Additionally, participants rated how often they use different types of social media platforms. Platforms primarily used for private purposes (e.g., Instagram, TikTok, Facebook) were used by 92.03% (n = 416) of participants at least once a day or more. In contrast, 60.84% (n = 275) of the participants reported never using social media platforms for professional purposes (e.g., LinkedIn, XING). 12.39% (n = 56) used them once or twice a month, and only 4.87% (n = 22) used them daily. The most frequently used platforms were messenger services such as WhatsApp or Telegram, with 98.46% (n = 445) reporting daily or more frequent use.

Participants were also asked to report their daily screen time, which could be checked on their smartphones. General screen time among participants showed substantial variation, with a mean of 251.94 min per day (*SD* = 153.55) and a median of 230 min. After excluding outliers, the mean screen time decreased slightly to 241.94 min (*SD* = 93.85, *Mdn* = 230). Screen time related to social media use showed a mean of 167.17 min (*SD* = 206.33, *Mdn* = 138) and a median of 138 min. Again, outliers were excluded and the screen time related to social media use declined to 147.46 min (*SD* = 75.02) and the median to 135 min of social media use per day. For a more detailed overview of participants’ use of social media platforms, see Appendix D.

### 4.2. Testing of Hypotheses

For all used scales, a composite mean score was computed, with higher values indicating greater levels of social media addiction, Fear of Missing Out, cyberloafing, work engagement, and organizational commitment. Descriptive statistics for the study variables are shown in Table 1.

To test Hypotheses 1–6, Pearson’s correlation coefficients (two-tailed, α = 0.05) were calculated. Hypothesis 1 examined the association between social media addiction and work engagement. The correlation was not significant (*r* = −0.08, *p* = 0.11). Therefore, H1 was not supported.

Hypothesis 2 examined the association between social media addiction and organizational commitment. Social media addiction was not significantly associated with organizational commitment (*r* = −0.03, *p* = 0.74). Therefore, H2 was rejected.

### 4.3. Network Analysis

An unregularized partial correlation network was estimated using the estimateNetwork function with the pcor argument in R, which computes pairwise partial correlations between all variables. This approach retains all edges regardless of strength and does not apply shrinkage or model selection. It should be noted that edges involving OC were based on the subsample of employed participants (n = 173), due to OC being measured only in this group. The resulting network is shown in Figure 1.

### 4.4. Network Structure

Among the five nodes and 10 edges of the network, two edges were notably strong. The nodes WE (“Work Engagement”) and OC (“Organizational Commitment”) had the strongest edge weight (r = 0.50). The edge weight between the nodes SMA (“Social Media Addiction”) and FOMO (“Fear of Missing Out”) was also very strong (r = 0.45). Additionally, moderate edge weights were found between OC and FOMO (r = −0.26) and SMA and CL (“Cyberloafing”; r = 0.24). Small edge weights were found between SMA and WE (r = −0.15), SMA and OC (r = 0.16), between FOMO and work engagement (r = 0.13), and between FOMO and CL (r = 0.10). Appendix E shows all the edge weights within the network. The bootstrapped 95% confidence interval was narrow and indicates that the estimation of edge weights was relatively precise (see Appendix E).

Nodes SMA (1.01) and OC (1.00) had the highest level of strength centrality, followed by FOMO (0.94) and work engagement (0.87). The lowest level of strength centrality was found in the node CL (0.52). Strength centrality showed limited stability (Correlation Stability (CS) coefficient *CS*(cor = 0.7) = 0.36) and should therefore be interpreted with caution. The stability of expected influence was acceptable, with a CS coefficient of *CS*(cor = 0.7) = 0.438, indicating that the stability of this centrality index is adequate for interpretation.

Social media addiction showed the highest expected influence (EI = 0.71), followed by work engagement (EI = 0.57) and Fear of Missing Out (EI = 0.43). In contrast, the lowest expected influence was found for cyberloafing (EI = 0.34) and organizational commitment (EI = 0.31). Although organizational commitment showed high strength centrality, its relatively low expected influence suggests that it is connected to several other nodes, but with a mix of positive and negative associations that may balance each other out. The closeness centrality values across nodes were relatively similar, ranging from CC = 0.03 to CC = 0.05. This indicates that all nodes in the network are comparably close to one another in terms of shortest path distances. However, the stability of closeness centrality was limited, with a correlation stability coefficient of *CS*(cor = 0.7) = 0.28. Therefore, interpretations based on closeness centrality should be made with caution. Regarding betweenness centrality, the stability coefficient was found to be *CS*(cor = 0.7) = 0.00, indicating insufficient stability. Consequently, betweenness centrality measures were excluded from further interpretation. Detailed centrality statistics, including both raw and z-standardized values, are presented in Table 2.

Two theoretical communities were defined for the bridge centrality analysis: Problematic Social Media Use (including Social Media Addiction, FoMO, and Cyberloafing) and Work-Related Attitudes (comprising Work Engagement and Organizational Commitment). Among the examined nodes, OC exhibits the highest Bridge Strength (BSC = 0.51), suggesting it plays a key role in linking the two communities, followed by FOMO (0.39) and WE (0.38). SMA (0.31) and CL (0.18) showed lower values. Bridge Expected Influence values, defined as an overall increase in node activation, are relatively close to zero or slightly negative, with WE (BEI = 0.07) and SMA (0.01) showing small positive influence, whereas FOMO (−0.12) and OC (−0.19) show slightly negative influence. This indicates that while some nodes have a small positive influence on bridging communities, others might exert less or an inhibitory influence in this context. The CS-coefficient for strength centrality (including bridge-related metrics) was moderate (0.44), suggesting reasonable but not optimal stability. Therefore, although interpretation of Bridge Strength and Bridge Expected Influence is informative, conclusions should be drawn cautiously.

### 4.5. Explorative Analyses: Network Analysis Using Graphical LASSO with EBIC Model Selection

Due to the large number of weak connections in the general network model, an additional network model was estimated using graphical LASSO regulization with EBIC model selection. To estimate the network structure, the graphical least absolute shrinkage and selection operator (glasso) was applied in combination with the extended Bayesian information criterion (EBIC; [32]). This approach is often used because it limits the number of spurious edges making the network more interpretable ([32]). This regulation process compresses edge weights and sets negligible partial correlations to zero ([32]). Therefore, the result is a more stable and sparser network. To identify the optimal network model, the tuning parameter of the EBIC was set to 0.5, which is commonly recommended to balance the sensitivity and specificity of detecting true edges ([36]).

The resulting network, which encompasses social media addiction, Fear of Missing Out, cyberloafing, work engagement, and organizational commitment among 452 university students, is illustrated in Figure 2. Note that partial correlations involving Organizational Commitment (OC) are based on the subset of employed participants (n = 173), since OC was only assessed in this group. Again, readers should consider the smaller sample when interpreting edges including OC. There were five nodes, but only four edges remained in the network. Nodes WE (Work Engagement) and OC (Organizational Commitment) had the strongest edge weight (*r*= 0.42). Nodes FOMO (Fear of Missing Out) and SMA (Social Media Addiction) also had strong edge weight (*r* = 0.39). Further, SMA and FOMO were both positively connected to the node CL (Cyberloafing) (SMA-CL: *r* = 0.20; FOMO-CL: *r* = 0.10). In the present study, the strongest relations (e.g., SMA–FOMO and WE–OC) emerged consistently across both networks.

Additionally, FOMO and OC maintained a negative edge weight of r = −0.13. The regularized network analysis revealed a triangular connection between the variables regarding problematic technology use and a strong positive connection between the two work-related variables. Interestingly, the only connection between both clusters was found between FOMO and OC. Appendix F summarizes the edge weights within the network. The stability and accuracy of the estimated edge weights in the regularized network were evaluated using a nonparametric bootstrap procedure with 1000 resamples. Most edge weights show acceptable robustness, whereas some connections have confidence intervals that include zero, indicating less stable estimations.

To further analyze the global structural characteristics of the regularized network, several network indices were computed. The network had a density of 0.5, suggesting that half of all possible connections between nodes were estimated to be nonzero after regularization. The network consisted of a single connected component, which indicates that all nodes were directly or indirectly related. Furthermore, the global clustering coefficient was 0.5, and the average local clustering coefficient was 0.47, suggesting that the tendency for nodes to form connected clusters is moderate. The stability of edge weights was found to be relatively high (CS edge = 0.60) and therefore indicates stable estimation of the structural connections of the network.

As for the first network, the highest strength centrality was found for nodes FOMO (0.63), SMA (0.59), and OC (0.51). The nodes WE (0.37) and CL (0.30), contrarily, seemed to be less important. Strength centrality again showed limited stability (CS strength = 0.36) and should be interpreted with caution. As seen in the first network, SMA showed in the second network again the highest expected influence (0.59), followed by WE (0.37), FOMO (0.36), CL (0.30), and OC (0.24). The correlation stability coefficient for expected influence (CS EI = 0.44) exhibited moderate stability and must be interpreted with caution. For betweenness (CS betweenness = 0.21) and closeness (CS closeness = 0.21), it revealed low stability; therefore, both measures were excluded from further interpretation. As was already apparent in the regularized network visualization, only FOMO and OC served as bridge nodes (both: EI = −0.13). Detailed centrality statistics for the EBICglasso network, including both raw and z-standardized values, are presented in Appendix F.

## 5. Discussion

### 5.1. Summary of Results and Interpretation

The aim of the study was to examine the interrelations of social media addiction, Fear of Missing Out, cyberloafing, work engagement, and organizational commitment. By doing so, the study addresses a critical research gap, as these variables are rarely investigated together. The findings contribute to a better understanding of how social media-related variables interact with work-related attitudes. The results revealed several crucial findings.

First, a triangular association between social media addiction, Fear of Missing Out, and cyberloafing became evident. As expected in Hypotheses 3 and 4, there was a significant positive association between social media addiction and Fear of Missing Out and between Fear of Missing Out and cyberloafing. These associations were confirmed both in the general network analysis and in the exploratory regularized network analysis. Furthermore, both networks revealed a positive association between social media addiction and cyberloafing. Taken together, these results highlight that problematic social media use does not manifest in isolation but rather in a cluster of behaviors. These findings are in line with previous research (e.g., [35]; [46]; [114]). What remains unclear, however, is how exactly the three variables interplay. According to [113] ([113]), FoMO predicts social media addiction because it elicits the urge to often check one’s own social network sites. In the study by [14] ([14]), FoMO was again identified as a predictor of social media addiction and had more impact than personality traits. In contrast, [49] ([49]) found evidence that social media addiction predicts FoMO because it may lead to an increased curiosity about what is going on at social media platforms. Therefore, the temporal and causal relationship between FoMO and social media use remains unclear ([45]).

Regarding the relationship between FoMO and cyberloafing, it is assumed that the constant worry of missing out on experiences that others have will, in turn, elicit cyberloafing behaviors to satisfy this need ([21]). Contrarily, [47] ([47]) were able to show that cyberloafing is associated with more FoMO. Similar findings are also found regarding the association of cyberloafing and social media addiction ([2]). Typically, participants prefer more pleasurable activities while procrastinating ([94]). Social media is often used for entertainment and passing time as well as for relaxation ([123]). Therefore, using social media is one possible activity that is performed while procrastinating. Accordingly, a positive association between procrastination and social media addiction was found ([69]). It is assumed that the urge to use social media platforms may lead to more cyberloafing behaviors ([104]).

To summarize, there is a lot of evidence that these three constructs strongly intercorrelate. Unfortunately, the direction of associations is not well understood so far. Besides, in research, these three variables are not often studied together. One possible explanation for the ambiguous relationships among the variables is that they may be mutually reinforcing, as suggested for the link between FoMO and social media addiction ([45]; [49]). These results are crucial, as they offer valuable insights into how social media addiction, FoMO, and cyberloafing can be addressed. It underlines the necessity of not examining these constructs in isolation but rather considering their interplay to better understand their impact, as well as to develop effective interventions.

Second, it is evident in the research conducted that FoMO and organizational commitment play a central role within the network. Both have high strength centrality and the highest bridge strength. Moreover, the association between organizational commitment and FoMO was the only bridging association that remained in the regularized model connecting the work-related attitudes with the variables describing social media use. As a result, the study contributes more strongly to understanding how motivational mechanisms connect these fields, rather than providing evidence for direct effects of social media addiction on organizational behavior. This finding is noteworthy because such an effect was not initially anticipated, and it challenges existing assumptions in the literature. It suggests that FoMO may play a much more central role in linking social media use with work-related attitudes than previously recognized, which calls for a reconsideration of how FoMO is conceptualized and studied in organizational research.

From a theoretical perspective, this result could be interpreted using the Self-Determination Theory ([26]). FoMO may reflect unmet needs for relatedness and autonomy, which in turn could affect organizational commitment ([68]; [93]). There is no research that focuses on the association between these two variables. Nonetheless, several researchers have argued that FoMO should be examined in a broader context ([27]; [90]). FoMO was found to be in the mediator role in various studies and therefore seems to be of high relevance in various fields ([27]; [90]). It is assumed that FoMO may play a role in the business context in terms of productivity, organizational commitment, job satisfaction, or leadership, among others ([27]; [90]). Thus, the findings of the present study reinforce this assumption and emphasize the significance of FoMO as a key factor in organizational contexts.

Third, there was only a small negative association between cyberloafing and both organizational commitment and work engagement in the general network analysis. This association was removed completely in the regularized network model. Hypotheses 5 and 6, which proposed these associations, were not significant and, in turn, were rejected. This finding is remarkable because it contradicts the common assumption in the literature that cyberloafing is inherently harmful to work-related attitudes. It contrasts with the majority of studies concerning cyberloafing, which tend to frame it mostly negatively (e.g., [78]). [88] ([88]), for example, revealed in their study a direct negative effect of work engagement on cyberloafing.

However, some recent literature suggests cyberloafing may not necessarily need to be discussed in a negative context but can also serve as a coping or recovery mechanism during work and lead to enhanced creativity and increased well-being ([67]; [71]). Taken together, this indicates that the relationship between cyberloafing and work outcomes may be more nuanced than assumed. These findings suggest that not all forms of cyberloafing are equal in nature or impact. While brief, conscious instances of cyberloafing (e.g., checking messages during a break) may help employees recover and enhance creativity, frequent or uncontrolled cyberloafing could indicate disengagement and harm productivity. Thus, future studies should distinguish between functional micro-breaks and persistent distraction behavior.

This is further supported by qualitative findings from [21] ([21]), who identified six themes in Millennials’ perceptions of cyberloafing, reflecting both its positive and negative facets. These included themes such as cyberloafing as a habit, cyberloafing as a result of boredom, or cyberloafing as a way to gain mental clarity and maintain productivity over a long period of time ([21]). [5] ([5]) found that people engage in cyberloafing behaviors at work to cope with stressful work situations. Cyberloafing can serve as a micro-break that is found to improve participants’ mood and reduce stress ([5]; [73]). Therefore, cyberloafing seems to have a more complex impact in the workplace than previously thought ([5]). These ambiguous findings regarding cyberloafing were further supported by a review by [112] ([112]). There, mixed results regarding the influence of cyberloafing on job performance were detected ([112]). These findings suggest that cyberloafing may have both positive and negative effects, outlining that especially the amount of time spent cyberloafing plays a crucial role in describing the effects ([112]). This underlines that cyberloafing should no longer be viewed as a multifaceted phenomenon with both potential risks and benefits for organizations.

A second aspect that should be considered when discussing cyberloafing is its increasing social acceptance in modern workplaces. [71] ([71]) found that employees considered cyberloafing behaviors at work as acceptable if it remained within a limit of 1 h and 15 min per day. This supports the idea that moderate cyberloafing within socially accepted boundaries may not be perceived as harmful. Consistent with this perspective, [53] ([53]) challenge the common classification of cyberloafing as counterproductive work behavior. Their findings indicate that cyberloafing tends to decrease when employees face a higher workload, suggesting that personal internet use at work often serves to fill unstructured or idle time rather than reflecting an intentional effort to avoid work responsibilities.

From this perspective, the way cyberloafing is typically described in academic literature might not accurately reflect its true nature. Rather than portraying it as a form of deviant behavior, it may be more accurate to interpret it as employees utilizing available time that has not been designated for specific tasks. However, exceeding such informal norms or using cyberloafing as a chronic avoidance strategy could still negatively impact work outcomes. Similarly, in the study by [72] ([72]) two common rationalizations to justify their cyberloafing behavior were detected: normalization and minimization. Normalization refers to the perception that everybody else is doing the same and it is therefore a common behavior ([72]). Minimization, on the other hand, describes the downplaying of the behavior’s impact on the organization, for instance by arguing that brief episodes of cyberloafing are harmless ([72]).

In sum, these findings suggest that the effects of cyberloafing may vary considerably across individuals and contexts. Rather than being inherently harmful, cyberloafing appears to serve diverse functions, which may even contribute to long-term productivity and well-being. Taken together, it is likely that cyberloafing only becomes problematic when it occurs excessively or serves as a persistent form of disengagement from work tasks. The literature suggests that cyberloafing should not be viewed as a uniform phenomenon, but rather as a spectrum ranging from functional to dysfunctional behavior. Functional cyberloafing refers to short, controlled activities that provide mental breaks and can support well-being and performance. In contrast, dysfunctional cyberloafing involves excessive, uncontrolled digital distractions that may indicate disengagement and reduce productivity. Recognizing this distinction is essential for both research and organizational practice.

Fourth, there was no significant association between social media addiction and both work engagement and organizational commitment. Therefore, Hypotheses 1 and 2 were rejected. Interestingly, in the general network, a weak negative association between social media addiction and work engagement and a weak positive association between social media addiction and organizational commitment was found, but both were removed in the regularized network. The removal of these associations in the regularized network suggests that the relationship may be confounded by other variables or spurious, underlining the importance of considering underlying mechanisms and contextual factors. According to the compensatory internet use theory ([60]), internet and social media addiction may serve as coping strategies for negative emotions rather than compulsive behaviors. Hence, social media addiction might result from stressful work situations without necessarily reducing job performance or positive work attitudes. Supporting this, [51] ([51]) found a weak positive link between social media addiction and organizational commitment.

These findings further support the assumption that social media addiction probably does not directly harm work-related attitudes.

Fifth, there was a strong positive association between work engagement and organizational commitment, which was stable both in the general network analysis and in the regularized network model. This finding is in line with previous research (e.g., [3]). According to the JD-R model, job resources can increase the motivation of employees, which can increase work engagement ([11]). It is assumed that job resources and work engagement serve as important predictors of organizational commitment ([48]). In this framework, organizational commitment can be seen as a consequence of high levels of work engagement ([105]; [102]). Notably, both organizational commitment ([80]; [96]) and work engagement ([24]) have been positively associated with job performance, highlighting their importance not only for employee attitudes but also for organizational outcomes.

### 5.2. Limitations

The present study has several limitations that should be considered. Similar to previous research, this study was conducted at universities with a young, university sample. Therefore, students with a part-time job in this study were allowed to decide on their own if they want to answer questions with regard to their study or with regard to their (part-time) job. Student participants without a job answered the questions with regard to their studies, respondents that were working full-time answered the questions with regard to their job. In line with [117] ([117]), this may result in unclear results because the setting in which the behavior takes place is different. Moreover, the consequences may differ heavily. For example, showing cyberloafing behavior at work may have much more serious consequences than cyberloafing behavior in a lecture, where you just need to sit and listen. Additionally, the experience of work engagement can be different at university or at work. Therefore, to obtain valid results for the professional context, it is recommended to examine full-time workers.

Another key limitation is that the study did not differentiate between functional and non-functional cyberloafing. As the measure used does not capture the distinction between functional and non-functional cyberloafing, the network may reflect an undifferentiated construct, reducing the clarity and strength of the conclusions and practical implications.

Furthermore, most publications concerning problematic social media use and addiction examine young students between 19 and 25 years of age ([92]). The same accounts for this study. Nowadays, the usage of social media is relevant for almost all generations and therefore the examination of addiction and correlates should be considered for a broader variety of age groups. The different age groups may express different attitudes and perceptions toward social media use and cyberloafing, which was already indicated by the study of [50] ([50]) regarding cyberloafing.

Another limitation concerns the broad and inconsistently defined nature of constructs such as social media addiction and problematic social media use. These terms can encompass a wide range of behaviors, from excessive but passive use to ethically questionable or harmful activities such as online stalking or cyberbullying ([111]). As a result, comparing findings across studies is difficult, and conclusions regarding their impact on work-related outcomes should be interpreted with caution.

### 5.3. Theoretical and Practical Implications

The present study provides new insights into problematic social media use at work and its association with work-related attitudes. A notable finding is the triangular relationship between cyberloafing, social media addiction, and Fear of Missing Out, indicating that these variables are strongly associated. For future studies, these associations should be considered and systematically examined together. Given the potential for these constructs to mutually reinforce one another, a kind of vicious cycle may emerge, which could help explain the inconsistencies and contradictions found in prior research regarding their causal relationships. Additionally, the findings highlight the central role of FoMO in linking problematic social media use with work-related attitudes. This suggests that FoMO may function as a psychological mechanism through which social media use impacts workplace behavior. By identifying FoMO as a potential bridge between these two major research areas, the present results challenge the notion of FoMO as solely a consequence of excessive social media use ([49]) and instead conceptualize it as a key commonality. On the other hand, this also illustrates a limitation of the study. The linkage of problematic social media use and work-related attitudes relies primarily on FoMO as the connecting construct rather than on direct relationships between the two research areas. This becomes particularly evident in the non-significant direct associations between social media addiction, organizational commitment, and work engagement. Instead of supporting the expected direct effects, the results suggest that FoMO may operate as the more relevant underlying process.

Furthermore, the ambiguous findings surrounding cyberloafing highlight the need for a more nuanced approach in organizational policies. It remains unclear whether each type of cyberloafing behavior is harmful or whether some can actually function as a micro-break that reduces work-related stress and increases well-being and job satisfaction ([5]). According to [15] ([15]), almost 90% of participants engaged in minor forms of cyberloafing, such as checking and sending emails and visiting websites. More serious forms of cyberloafing, including using chatrooms or visiting websites for online gambling, occurred much less frequently ([15]). [97] ([97]) demonstrated that especially personal downloading, online shopping, and personal web browsing lead to inefficiency, which subsequently decreases productivity and ultimately lowers job performance ([97]). In his study, up to 25% of work time was wasted unproductively ([97]). These more time-consuming and non-work-related activities exemplify the dysfunctional end of the cyberloafing spectrum, contrasting with minor behaviors such as brief messaging, which may serve recreational purposes. This may explain why findings regarding cyberloafing vary so widely and suggest that more serious forms of cyberloafing are particularly problematic and should be addressed through organizational interventions. Therefore, organizations should avoid treating all cyberloafing behaviors as counterproductive. Instead, policies and organizational interventions should differentiate between short, recovery-oriented online breaks (functional) and more disruptive, task-avoidant behaviors (non-functional).

From a practical standpoint, the findings further suggest that organizations should not only pay attention to employees’ private social media use at work but also to their motives and psychological experiences related to it, especially Fear of Missing Out. As was found in the present study, FoMO appears to play a central role in linking problematic social media use to work-related attitudes and behaviors and may function as a key psychological mechanism that explains how social media engagement relates to workplace outcomes. Organizational interventions should focus on reducing FoMO as a potential driver of problematic social media use, rather than targeting solely cyberloafing behavior.

### 5.4. Future Research Directions

Future research should investigate the relationship between internet addiction and cyberloafing behaviors in more detail. Both terms are best understood as broad categories that encompass a wide range of distinct phenomena. Of particular relevance to both is the concept of smartphone addiction. With the shift in social media use from desktop computers and laptops to smartphones, the smartphone has become the primary device for social media engagement ([59]). [29] ([29]) found that participants who owned a smartphone scored significantly higher on cyberloafing than those who did not. Similarly, [91] ([91]) identified a significant positive association between smartphone addiction and cyberloafing behavior among students. Moreover, a study by [50] ([50]) revealed that more than 50% of participants reported accessing the internet via their smartphones or personal laptops at work. Mobile social networking app usage has been identified as a significant predictor of mobile addiction ([103]). Smartphone addiction may therefore be understood as a subcomponent of both social networking site addiction ([66]) and internet addiction ([91]). In addition, [29] ([29]) demonstrated a significant association between cyberloafing and the time spent on various social media platforms. Taken together, these findings suggest that smartphone addiction plays a key role in understanding cyberloafing, social media addiction, and related phenomena. However, the present study did not distinguish between different forms of technology use, such as smartphone- or computer-based social media access, nor did it account for nuanced differences between internet addiction, smartphone addiction, and social media addiction.

A second aspect that is interesting for future research is to examine different types of social media platforms and their specific characteristics. Unfortunately, existing definitions of social media are often inconsistent and do not capture the nuanced differences across platforms ([22]). In line with [92] ([92]), it needs to be considered that social media platforms differ in terms of functionality, algorithmic structures, and recommendation systems. Due to the distinct features, users are motivated by various reasons for using a specific platform and therefore behave differently when using it ([92]). This indicates that social media addiction should not be viewed as a homogeneous phenomenon, as its characteristics and impact may vary by platform. Most studies in this field have focused on Facebook and Instagram, but newer platforms such as TikTok or platforms with different purposes, such as YouTube or WhatsApp, are rapidly gaining popularity and require closer investigation ([92]).

A third aspect that may be interesting for future research is the development of intervention strategies. In research, especially interventions for social media addiction are evaluated ([55]). There are interventions such as cognitive reconstruction or self-help interventions through using applications ([55]). Furthermore, web-based interventions using features like automatic notifications, usage limits, and reward systems have demonstrated potential in reducing Facebook addiction among postgraduate students ([28]). With the present study highlighting the importance of FoMO, the question arises if these interventions also are applicable for reducing FoMO through lowering social media use or if it is necessary to address this underlying Fear of Missing Out on something by a different approach. There is some evidence in research that perceived social support may be a key factor in experiencing FoMO ([39]). Low social support contributes to unmet needs for belonging, which in turn may increase feelings of FoMO ([39]). Additionally, there is evidence that also factors such as stress perception ([37]) and the need for social acceptance ([68]) play an important role. Future studies should investigate whether interventions targeting social media addiction are also suitable for reducing FoMO or whether distinct psychological drivers, such as low social support or stress, require different intervention approaches.

A final consideration for future research concerns the cultural context of existing studies. During the literature review, it became apparent that a substantial amount of empirical work on social media addiction, Fear of Missing Out, and cyberloafing originates from non-Western countries, particularly from Turkey as well as East and Southeast Asia (e.g., China, South Korea, and Malaysia). There, cultural norms, work environments, and patterns of social media use may differ significantly from those in Western contexts. For example, comparative research has found that social media addiction levels among university students in South Korea are notably higher than those in Turkey, with South Korean students reporting greater emotional support from social media but also more conflicts and negative consequences related to their use ([116]). This raises concerns about the generalizability of findings. Future research should therefore aim to replicate and extend these findings in more diverse cultural settings to examine whether the identified mechanisms hold across different socio-cultural environments or whether cultural variables may moderate these relationships.

## 6. Conclusions

This study focuses on problematic social media use at work by applying a network analytical approach. A strong and stable relationship between work engagement and organizational commitment provides support for established models such as the JD-R model. Fear of Missing Out emerged as a central mechanism linking social media addiction and work-related attitudes. The triangular link between FoMO, social media addiction, and cyberloafing may represent a self-reinforcing cycle contributing to persistent dysfunctional behavior.

Interestingly, social media addiction showed only weak ties to work attitudes, highlighting the importance of linkage through variables like FoMO or cyberloafing. From a practical perspective, interventions should address not only screen time but also the psychological drivers of use. Differentiating between types of cyberloafing and platform-specific behaviors could further inform targeted strategies.

## Figures and Tables

**Figure 1 behavsci-15-01719-f001:**
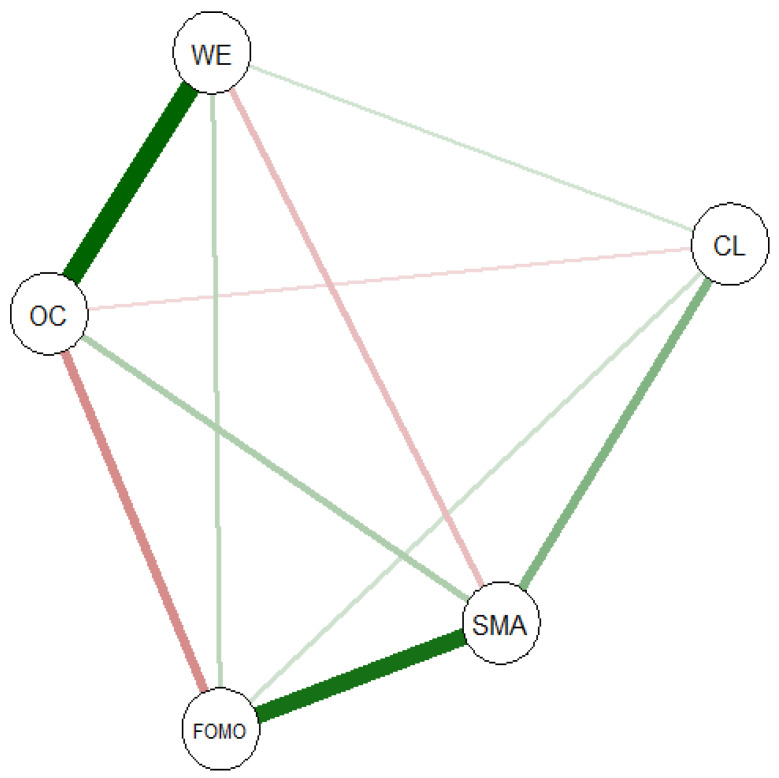
Network structure based on partial correlations. Abbreviations: SMA = Social media addiction, FOMO = Fear of Missing Out, CL = Cyberloafing, WE = Work Engagement, OC = Organizational Commitment; thicker lines mean stronger correlations.

**Figure 2 behavsci-15-01719-f002:**
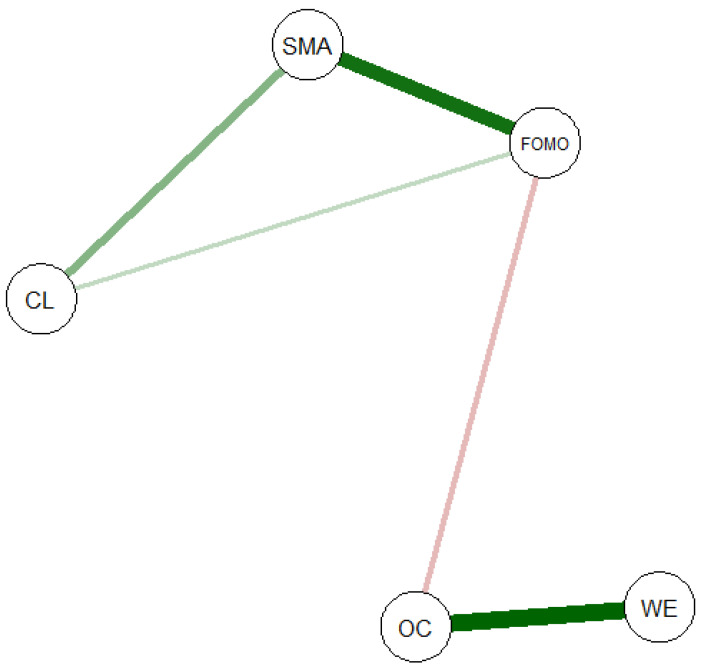
Network structure based on the EBICglasso estimation. Abbreviations: SMA = Social media addiction, FOMO = Fear of Missing Out, CL = Cyberloafing, WE = Work Engagement, OC = Organizational Commitment. Thicker lines mean stronger correlations.

**Table 1 behavsci-15-01719-t001:** Descriptive Statistics, Response Formats, and Reliability Estimates of the Study Variables.

Variable/Scale	Number of Items	Response Format	Cronbach’s α	*M*	*SD*
Bergen Social Media Addiction Scale	6	Very rarely (1)–very often (5)	0.79	2.68	0.80
Fear of Missing Out Scale	10	Not at all true of me (1)–Extremely true of me (5)	0.74	2.62	0.59
Cyberloafing Scale	19	Never (1)–Once a day (4)–Constantly (6)	0.89	2.39	0.73
Utrecht Work Engagement Scale	9	Never (1)–Always (7)	0.92	4.21	0.99
Organizational Commitment Questionnaire (OCQ)	15	Totally disagree (1)–Totally agree (5)	0.91	3.20	0.78

**Table 2 behavsci-15-01719-t002:** Centrality statistics across all nodes.

	Node	Measure	Value (Absolute)	Value (z-Standardized)
1	Social Media Addiction	Betweenness	3	1.38
2	Fear of Missing Out	Betweenness	2	0.61
3	Cyberloafing	Betweenness	0	−0.92
4	Work Engagement	Betweenness	0	−0.92
5	Organizational Commitment	Betweenness	1	−0.15
6	Social Media Addiction	Closeness	0.05	0.83
7	Fear of Missing Out	Closeness	0.05	1.06
8	Cyberloafing	Closeness	0.03	−1.37
9	Work Engagement	Closeness	0.03	−0.55
10	Organizational Commitment	Closeness	0.05	0.04
11	Social Media Addiction	Strength	1.01	0.69
12	Fear of Missing Out	Strength	0.94	0.36
13	Cyberloafing	Strength	0.52	−1.72
14	Work Engagement	Strength	0.87	0.02
15	Organizational Commitment	Strength	1.00	0.66
16	Social Media Addiction	Expected Influence	0.71	1.42
17	Fear of Missing Out	Expected Influence	0.43	−0.27
18	Cyberloafing	Expected Influence	0.34	−0.78
19	Work Engagement	Expected Influence	0.57	0.60
20	Organizational Commitment	Expected Influence	0.31	−0.97

## Data Availability

The dataset generated and analyzed during the current study is available in the OSF repository: https://osf.io/um7wc/resources, accessed on 1 December 2025.

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
