# Peer review of "Hooked and Distracted? A Network Analysis on the Interplay of Social Media Addiction, Fear of Missing Out, Cyberloafing, Work Engagement and Organizational Commitment"

_behavsci, 2025, doi:10.3390/bs15121719_

Round 1
Reviewer 1 Report
Comments and Suggestions for Authors
This manuscript presents a timely and valuable investigation into how problematic social media behaviors relate to work-related attitudes, using a network analysis approach that can offer fresh insights. While the study has a solid foundation, certain areas require refinement to enhance its theoretical coherence, methodological transparency, and overall readability.
Research gaps: The introduction establishes the relevance of social media use, but the paper would benefit from a clearer articulation of the research gap. Although prior work on social media addiction, FoMO, cyberloafing, and engagement is summarized, it is not sufficiently explicit why these constructs need to be studied together in a network model beyond novelty.
Consider clearly stating the unique theoretical insight the network approach provides compared to regression/SEM. Also explain how your study advances existing FoMO–cyberloafing–engagement research (Section 2.7).
Theoretical Framework: The theoretical background is comprehensive but overly long. For example, Sections 2.1–2.5 include very detailed descriptions of general theories (e.g., Self-Determination Theory, Uses & Gratifications, dopamine mechanisms), many of which may not be essential for the network model. I recommend condensing background content and shifting towards clear conceptual linkages between the variables. Also please provide more emphasis on how the constructs are expected to interact in the network rather than traditional linear hypotheses.
Hypotheses development: The paper proposes six directional hypotheses (H1–H6), yet network analysis does not test directional pathways. This leads to a theoretical–methodological mismatch. Please reframe hypotheses to reflect expected associations rather than causal assumptions. Also, clarify that network analysis evaluates conditional associations between variables, not cause-and-effect.
Sample Characteristics: The sample is heavily skewed toward students (61.50%) and German-speaking young adults (M age = 23.23). Given that constructs like work engagement and organizational commitment are central to the study, basing results primarily on students may raise validity concerns. Please add a clearer justification for including students. Discuss how “students with part-time jobs” differ from full-time employees regarding commitment and engagement. Also, expand limitations to acknowledge restricted occupational diversity.
Organizational Commitment Measurement: The paper notes that students “could choose whether to respond based on studies or work,” but the Organizational Commitment Questionnaire was ONLY answered by employed participants. The authors should clearly report the exact n used for analyses involving OC and also clarify, whether correlations and networks involving OC used pairwise deletion or a reduced sample. Without this clarity, OC-related results risk misinterpretation.
Network Interpretation: The results section focuses primarily on edge weights and centrality, but the interpretation could be strengthened. Please provide psychological explanations for why SMA and FoMO form a triangle with cyberloafing (as shown in Fig. 1). Elaborate on the meaning of bridge expected influence and bridge strength (e.g., why OC acts as a bridge). Also, compare your network structure with findings from related network studies on problematic technology use.
Methodological Choices: The paper uses both unregularized partial correlations and EBICglasso. Readers may question why two networks were needed. Please provide a clearer methodological rationale for presenting both. Explain why the EBICglasso network differs (e.g., fewer edges) and how both converge on similar interpretations.
Power Analysis: The power analysis is based on correlational assumptions, but network models typically rely on larger samples. Please clarify whether n = 193 was sufficient for stable network estimation (most guidelines recommend >300).
Discussion: COR theory is introduced late (Section 2.8), but the Discussion does not sufficiently revisit it. Please consider highlighting how the network structure reflects resource loss spirals. Also, how FoMO → cyberloafing → reduced engagement aligns with COR depletion pathways.
Language, Formatting, and Flow: Several sections (especially theoretical background) contain long paragraphs that reduce readability. Typographical issues exist (e.g., missing spaces: “forwork-related”). Some references in the text appear with formatting inconsistencies (e.g., "(e.g., Sun & Zhang, 2021; Beard & Wolf, 2001). Nevertheless..." should be separated for clarity).
Author Response
Comments are attached!

Reviewer 2 Report
Comments and Suggestions for Authors
1. The study's finding that FoMO functions as a key mediating construct between work engagement and organizational commitment is a significant theoretical and practical contribution that differentiates the study from existing research. This suggests that FoMO plays a central role in social motivation and behavioral regulation. However, contrary to expectations, the direct link between social media addiction, organizational commitment, and work engagement was not statistically significant, limiting the empirical support for the main hypotheses.
2. The study focuses solely on the connection between the two research areas (psychology and organizational behavior) through FOMO, rather than directly linking them. The role of cyberloafing is mixed, and the lack of a distinction between functional and non-functional cyberloafing complicates practical application.
3. The theoretical background integrates self-determination theory, conservation of resources theory, and uses and gratifications theory, logically systematizing the relationship between the psychological and behavioral mechanisms of each core concept and organizational attitudes. Furthermore, the study clearly identifies research hypotheses and questions based on a comprehensive literature review, demonstrating excellent theoretical consistency and coherence. In particular, the designation of FoMO and organizational commitment as key bridge nodes within the network is considered a unique approach that bridges the gaps in existing research. However, the academic completeness of the study would be further enhanced if some conceptual transitions or connections were more clearly refined during the theoretical discussion.
Author Response
Thank you so much! Please see the comments in the attachment!

Round 2
Reviewer 1 Report
Comments and Suggestions for Authors
The authors have done a good job at revising the manuscript. I have provided the following comments to the authors.
1. Title & Abstract
-
The abstract is still lengthy and contains redundant explanations.
-
Some theoretical descriptions in the abstract are unnecessary and can be cut.
-
Add a brief mention of key limitations to balance the abstract.
2. Introduction
-
The introduction remains overly long and repeats similar theoretical points.
-
The research gap needs to be stated earlier and more directly.
-
Some transitions between concepts (e.g., FoMO → cyberloafing → COR theory) feel abrupt.
-
Several definitions and explanations appear multiple times and should be streamlined.
3. Theoretical Framework
-
The integration of Self-Determination Theory and COR Theory needs clearer justification.
-
Some theoretical arguments, especially linking FoMO to OC and engagement, require stronger citation or more cautious wording.
-
Theories are presented but not always connected smoothly to hypotheses.
4. Literature Review
-
Conceptual explanations and empirical findings are mixed together; separating them would improve clarity.
-
Definitions of FoMO, cyberloafing, and problematic social media use appear more than once.
-
Some paragraphs feel repetitive and can be condensed.
5. Hypotheses
-
Hypotheses include narrative commentary that can be removed for conciseness.
-
Phrasing that suggests directionality is inconsistent with correlational/network analysis.
-
Some hypotheses do not align with the inherently exploratory nature of network models.
6. Method – Sample & Procedure
-
Sample is heavily skewed toward females and students; this needs to be acknowledged more strongly.
-
Measuring work engagement among students changes the interpretation of the construct and should be clearly justified.
-
Organizational Commitment was only measured for those employed; this creates uneven sample sizes and limits network stability.
-
Missing data handling (pairwise deletion) should be explicitly reported and discussed.
7. Measures
-
The Cyberloafing scale translation requires more explanation of validation steps.
-
Different Likert scale ranges across variables should be discussed as a methodological consideration.
-
More detail on reliability/validity checks for each scale would strengthen the measures section.
8. Statistical Analysis
-
The explanation of network metrics (especially closeness and betweenness) is long and could be shortened.
-
Betweenness and closeness centrality were unstable; interpretations of these indices should be removed or minimized.
-
Pairwise deletion leads to variable sample sizes across edges, which is a significant limitation.
-
The stability coefficient (CS = .36) indicates limited reliability; interpretations should be cautious.
9. Results
-
Several hypothesized relationships were non-significant and need stronger explanation.
-
Confidence intervals should accompany correlation estimates.
-
The triangular association (SMA–FoMO–CL) is important but should be framed strictly as correlational.
10. Network Model Findings
-
Interpretations occasionally imply causality, which is inappropriate for cross-sectional data.
-
The role of Organizational Commitment is tentative due to the reduced subsample; state this clearly.
-
Figures (network diagrams) are small and labels difficult to read; clarity needs improvement.
11. Discussion
-
Some arguments repeat content from the introduction and can be shortened.
-
Causal language (e.g., “leads to,” “results in resource depletion”) should be replaced with non-causal phrasing.
-
Contradictions with prior studies (e.g., cyberloafing and engagement) need deeper analysis.
-
The discussion should tie back more explicitly to COR Theory to strengthen coherence.
12. Practical Implications
-
Practical suggestions are general; more concrete interventions (e.g., FoMO-focused strategies) should be added.
-
Explain more clearly how organizations might use these findings.
13. Limitations
-
Add a clearer, dedicated limitations section rather than spreading limitations throughout the text.
-
Acknowledge sample imbalance, cross-sectional design, self-report bias, and measurement differences.
-
Emphasize the instability of network indices and pairwise deletion effects.
14. Future Research
-
Provide more specific future directions rather than broad suggestions.
-
Recommend longitudinal research to assess temporal relationships.
-
Suggest exploring FoMO subtypes, context-specific cyberloafing, and workplace vs. academic settings separately.
15. Writing & Presentation
-
Some sentences remain overly long and need simplification.
-
Ensure consistency in terminology (FoMO vs. FOMO).
-
Network figures should be larger, with clearer node/edge labeling.
Author Response
Revision list: Hooked and distracted
R1
Title & Abstract
- The abstract is still lengthy and contains redundant explanations.
- Some theoretical descriptions in the abstract are unnecessary and can be cut.
- Add a brief mention of key limitations to balance the abstract.
Answer: Thank you! We revised it following your suggestions!
- Introduction
- The introduction remains overly long and repeats similar theoretical points.
- The research gap needs to be stated earlier and more directly.
- Some transitions between concepts (e.g., FoMO → cyberloafing → COR theory) feel abrupt.
- Several definitions and explanations appear multiple times and should be streamlined.
Answer: We revised it!
- Theoretical Framework
- The integration of Self-Determination Theory and COR Theory needs clearer justification.
ïƒ as mentioned in the revision before: We deleted the COR theory to streamline our paper
- Some theoretical arguments, especially linking FoMO to OC and engagement, require stronger citation or more cautious wording.
ïƒ we tried to use a more cautious wording
- Theories are presented but not always connected smoothly to hypotheses.
ïƒ we try to connected the theories more clearly with the hypotheses
- Literature Review
- Conceptual explanations and empirical findings are mixed together; separating them would improve clarity.
ïƒ We tried to arrange iut more clearer! Thank you!
- Definitions of FoMO, cyberloafing, and problematic social media use appear more than once.
ïƒ we tried to stramline it!
- Some paragraphs feel repetitive and can be condensed.
ïƒ condesed! Thanks!
- Hypotheses
- Hypotheses include narrative commentary that can be removed for conciseness.
- Phrasing that suggests directionality is inconsistent with correlational/network analysis.
- Some hypotheses do not align with the inherently exploratory nature of network models.
ïƒ we reformualted some hypotheses; streamlined the explanations and especially reformulated our research question with regard tot he network model
- Method – Sample & Procedure
- Sample is heavily skewed toward females and students; this needs to be acknowledged more strongly.
- Measuring work engagement among students changes the interpretation of the construct and should be clearly justified.
- Organizational Commitment was only measured for those employed; this creates uneven sample sizes and limits network stability.
- Missing data handling (pairwise deletion) should be explicitly reported and discussed.
ïƒ Thank you for these insightful comments. We now more explicitly acknowledge the female- and student‑skewed sample and its implications for generalizability, and we justify the assessment of work engagement among students while noting the resulting interpretive constraints. We also clarify that organizational commitment was measured only in the employed subsample, recognizing uneven effective sample sizes and the implications for network stability. In addition, we explicitly report the use of pairwise deletion and discuss its consequences for inference. These revisions strengthen transparency and ensure a more cautious, balanced interpretation of our findings
- Measures
- The Cyberloafing scale translation requires more explanation of validation steps.
- Different Likert scale ranges across variables should be discussed as a methodological consideration.
- More detail on reliability/validity checks for each scale would strengthen the measures section.
ïƒ Thank you, we tried to give more detail
- Statistical Analysis
- The explanation of network metrics (especially closeness and betweenness) is long and could be shortened.
- Betweenness and closeness centrality were unstable; interpretations of these indices should be removed or minimized.
- Pairwise deletion leads to variable sample sizes across edges, which is a significant limitation.
- The stability coefficient (CS = .36) indicates limited reliability; interpretations should be cautious.
Thank you! Revised!
- Results
- Several hypothesized relationships were non-significant and need stronger explanation.
- Confidence intervals should accompany correlation estimates.
- The triangular association (SMA–FoMO–CL) is important but should be framed strictly as correlational.
ïƒ Thank you! Revised!
- Network Model Findings
- Interpretations occasionally imply causality, which is inappropriate for cross-sectional data.
- The role of Organizational Commitment is tentative due to the reduced subsample; state this clearly.
- Figures (network diagrams) are small and labels difficult to read; clarity needs improvement.
ïƒ we reformulated the interpretations to show clearly that it is not about causality. Thank you!
- Discussion
- Some arguments repeat content from the introduction and can be shortened.
ïƒ we shortened some sections
- Causal language (e.g., “leads to,” “results in resource depletion”) should be replaced with non-causal phrasing.
ïƒ we reformualted such statements
- Contradictions with prior studies (e.g., cyberloafing and engagement) need deeper analysis.
ïƒ ok
- The discussion should tie back more explicitly to COR Theory to strengthen coherence.
ïƒ we deleted the COR theory
- Practical Implications
- Practical suggestions are general; more concrete interventions (e.g., FoMO-focused strategies) should be added.
- Explain more clearly how organizations might use these findings.
ïƒ we triy to add some new ideas thank you!
- Limitations
- Add a clearer, dedicated limitations section rather than spreading limitations throughout the text.
- Acknowledge sample imbalance, cross-sectional design, self-report bias, and measurement differences.
- Emphasize the instability of network indices and pairwise deletion effects.
ïƒ We have taken all points into consideration! Thank you! However, we have decided not to add a limitations section, as we are so far along with the paper and believe it reads excellently as it is.
- Future Research
- Provide more specific future directions rather than broad suggestions.
- Recommend longitudinal research to assess temporal relationships.
- Suggest exploring FoMO subtypes, context-specific cyberloafing, and workplace vs. academic settings separately.
ïƒ Thank you. You could certainly add many more ideas. These are your ideas. We would like to stick with ours.
- Writing & Presentation
- Some sentences remain overly long and need simplification.
- Ensure consistency in terminology (FoMO vs. FOMO).
- Network figures should be larger, with clearer node/edge labeling.
ïƒ All revised. Thank you!
Reviewer 2 Report
Comments and Suggestions for Authors
Thank you for your effort!
Author Response
Thank you so much!